# Homogeneous Sensor Fusion Optimization for Low-Cost Inertial Sensors

**DOI:** 10.3390/s23146431

**Published:** 2023-07-15

**Authors:** Dusan Nemec, Jan Andel, Vojtech Simak, Jozef Hrbcek

**Affiliations:** Department of Control and Information Systems, Faculty of Electrical Engineering and Information Technology, University of Žilina, 01026 Zilina, Slovakia; jan.andel@uniza.sk (J.A.); vojtech.simak@uniza.sk (V.S.); jozef.hrbcek@uniza.sk (J.H.)

**Keywords:** calibration, data averaging, MEMS gyroscope, sensor fusion

## Abstract

The article deals with sensor fusion and real-time calibration in a homogeneous inertial sensor array. The proposed method allows for both estimating the sensors’ calibration constants (i.e., gain and bias) in real-time and automatically suppressing degraded sensors while keeping the overall precision of the estimation. The weight of the sensor is adaptively adjusted according to the RMSE concerning the weighted average of all sensors. The estimated angular velocity was compared with a reference (ground truth) value obtained using a tactical-grade fiber-optic gyroscope. We have experimented with low-cost MEMS gyroscopes, but the proposed method can be applied to basically any sensor array.

## 1. Introduction

MEMS inertial sensors (i.e., gyroscopes and accelerometers) allow for the development of low-cost robotic and other mobile applications, where high-end fiber-optic gyroscopes (FOGs) are not affordable. The greatest concern is the bias stability and the noise of the MEMS sensors, which are much more significant compared to the FOG. Several approaches have been developed to compensate for the miscalibration and noise of the MEMS sensors. We may divide them into two categories: calibration and redundancy.

### 1.1. Calibration

The aim is to calculate the transformation function between the sensor output (raw data) and the best estimate of the measured variable (e.g., angular velocity, in the gyroscope case). The calibration is performed sample-by-sample and may also consider the sensor’s gain, bias, misalignment, and optionally higher-order nonlinearity. Especially in the case of the MEMS sensors, the calibration parameters change after the sensor restarts [1] and also significantly depend on the temperature [2]. There are two types of calibration applicable to the MEMS gyroscopes:Offline—the sensor is rotated by several predefined angular velocities. The measured points are then fitted using a calibration curve. We need to measure all angular velocities at different temperatures to compensate for the thermal drift. The most basic version of this is the start-up bias calibration, which computes the gyroscope’s bias (offset) by measuring the steady state’s mean output during the start-up phase while neglecting the Earth’s rotation. The authors of [3] applied a convolution neural network (CNN) to obtain the calibration model. When a simple linear calibration model is used, its coefficients can be calculated using the least squares method. The external stimuli may be, in exceptional cases, replaced with internal ones, e.g., in the case of honeycomb disk resonator gyroscopes (HDRG), the authors of [4] analyzed the third-order harmonic component of the sensor signal to estimate the scale factor of the closed-loop sensor, as well as to compensate for its thermal drift. The authors of [5] improved the precision of the calibration procedure by detecting the outliers in the measured calibration data using the random sample consensus algorithm (RANSAC), Mahalanobis distance, and median absolute deviation. A significant source of systematic errors is the misalignment of the sensor. The authors of [6] applied correlation analysis and the Kalman filter to estimate the installation errors.Online—the sensor readings are compared with the readings of other sensors (e.g., in the sensor array [7,8,9]) and/or with the previous readings of the same sensor (see, e.g., [10]). Such methods overcome the static nature of the offline calibration, compensating both long-term and short-term drift in the parameters using the Kalman filter or an algebraic estimator combined with a finite-response (FIR) filter [11,12,13]. An essential advantage of such methods is avoiding the requirement of multipoint offline thermal calibration since they adjust the calibration constants in real time.

### 1.2. Redundancy

The combination (fusion) of the information obtained from multiple sensors (possibly of different types) has the potential to be more precise, robust, and reliable compared to a single sensor [14]. In a standard configuration, the MEMS gyroscopes are coupled with the MEMS accelerometers (often in the same chip). The accelerometer estimates the vertical direction by measuring the gravity acceleration. The accelerometer is also inherently affected by the linear acceleration of the object itself concerning the inertial frame of reference. Still, in many applications, the long-term mean of the linear acceleration can be considered negligible concerning the gravity acceleration. The estimated vertical direction from the accelerometer can be combined with the gyroscope readings using various sensor fusion methods, improving the precision of the estimated angular velocity and Euler angles (see, e.g., [8,9,15,16,17,18,19]). Using multiple sensors also improves the safety of the whole system (see, e.g., [20,21]). To estimate the position of the object, the accelerometer and gyroscope readings can be combined with the odometrical data [22], microwave range finder [23], or laser scanner [24]. A variation of the nonlinear Kalman filter (e.g., the extended Kalman filter or EKF) is usually applied as the maximum likelihood estimator. The advantage of sensor fusion is its ability to estimate some parameters of the environment. Researchers in [25] used multiple independent models to enhance the robustness of an INS-based navigation system combined with a Doppler velocity log (DVL). The EKF requires presetting covariance matrices for the measurement. In real-life scenarios, the noise characteristics of the sensors are not constant; hence, they need to be estimated in real time. One commonly used technique is applying an LSTM neural network to learn how to estimate those changing parameters from the past readings of the sensors. For example, the authors of [11] developed a self-learning square-root cubature Kalman filter based on LSTM networks for GPS/INS navigation systems. However, estimators based on the neural networks are poorly explainable in general and hence cannot be used in safety-related applications. The fusion of the homogeneous sensor array can be implemented as a linear minimum variance (LVM) estimation (see, e.g., [26]). Such an approach requires one to know or precalculate the error covariance matrix, which is the same drawback that the EKF has.

### 1.3. Calibration in the Redundant Sensor Array

This article focuses on combining the approaches mentioned above, especially in cases where multiple inertial sensors of the same type are being used. Theoretically, when systematical errors of individual sensors are eliminated, the only source of error is the random (unpredictable) noise. Researchers in [8] investigated the precision of a MEMS array comprising 16 IMUs when the systematical errors were suppressed by offline calibration. It is more desirable to calculate each sensor’s deviation for the overall estimate of the measured variable. The authors of [27] proposed a method for online calibration for the array of 32 IMUs (later reduced to 8 IMUs due to the low communication bandwidth available) using a maximum likelihood estimator, compensating bias, gain, and misalignment errors. The reference value of the angular velocity has been estimated from the rotation of the measured gravity acceleration vector. Such a method is applicable only when vibrations and linear acceleration are not present (e.g., rotating the IMU array by hand as suggested by authors).

An essential part of evaluating the algorithm’s precision is the reference measurement system. The standard approach is to use a turntable. Researchers in [15] used military grade AHRS (Attitude and Heading Reference System) STIM300 to provide the reference value of angular velocity, compared with the values measured by a redundant array of 6 IMU units in a cubic constellation. Our first experiments used a rotational platform driven by a stepper motor, but it caused irreducible rotational-mode vibrations, rendering the reference value non-usable. Later, we used navigation-grade IMU SPAN-CPT for measurement of the ground-truth angular velocity.

## 2. Error Model of the Inertial Sensor System

### 2.1. Random Errors

Each sensor generates noisy readings. In the ideal case, we may assume that the noise is superposed to the true value. For the gyroscope, the relation is:(1)$$\omega ∼\omega_{\mathrm{true}}+\nu ,$$
where $\omega_{\mathrm{true}}$ is the true angular velocity, ω is the raw analog value at the output of the MEMS sensor unit, and ν is the noise from normal (gaussian) distribution:(2)$$\nu ∼N(0,\omega_{\mathrm{rms}}{}^{2}).$$

With only one sensor, the mechanical vibrations of the rotational character may also be considered noise.

### 2.2. Systematic Errors

The raw value of the gyroscope is truncated to the sensor’s full-scale range ±$\omega_{\max}$ and quantized to obtain the digital value. The quantization is a rounding of the analog value towards the nearest digital level and is performed by an internal A/D converter built-in within many commercially available MEMS sensors. The digital raw value converted to SI units (International System of Units) can be modeled by the following:(3)$$\omega_{\mathrm{raw}}=\Delta \omega \cdot \mathrm{round}\left(\frac{\omega_{\mathrm{truncated}}}{\Delta \omega }\right)=\Delta \omega \cdot \left\|\frac{\min\left(\max(\omega ,\;-\omega_{\max}),\;\omega_{\max}\right)}{\Delta \omega }\right\|.$$
where Δω is the quantization step (assuming uniform quantization), $\omega_{\mathrm{truncated}}$ is the analog raw value (true angular velocity + random noise) truncated within the sensor’s full-scale range, and round(*x*) is the rounding function returning the nearest integer to *x*, further noted as ‖x‖. All variables in Equation (3) are in rad·s^−1^. The quantization step is:(4)$$\Delta \omega =\frac{\omega_{\max}}{2^{r-1}},$$
where *r* is the sensor’s resolution in bits (e.g., *r* = 16 bits for the MPU9250 sensor). The histogram of the real gyroscope noise is shown in Figure 1.

Since the gyroscope is a directional sensor (measures angular velocity around one or multiple principal perpendicular axes), the misalignment of the sensor is a source of the systematic error:(5)$$\omega ∼\omega_{\mathrm{true}}\cos\varepsilon ,$$
where *ε* is the misalignment angle. In the case of three-dimensional movements, the misalignment causes cross-talk between axes, which is expressed by the rotational matrix ***R***_align_:(6)$$\omega ∼R_{\mathrm{align}}\cdot \omega_{\mathrm{true}}.$$

Another source of systematic errors is the miscalibration of the sensor—deviation of its gain and/or offset:(7)$$\omega ∼G\omega_{\mathrm{true}}+B,$$
where *G* is the gain and *B* is the bias. The gain and bias are generally unpredictable and vary with time and temperature. The MEMS gyroscope measures the Coriolis force affecting periodically oscillating mass. Vibrations of the measured object perpendicular to the sensing axis of the gyroscope may cause additional periodic components of the measured signal due to the resonance. Such disturbance can be modeled using discrete Fourier series. It is a sum of harmonic (sinus) signals with frequencies from 0 to Nyquist frequency $F_{s}/2=1/(2T_{s})$. The smallest frequency change detectable by the sensor with the constant sampling frequency $F_{s}$ is $F_{s}/N$, where N is the count of samples (window size). The model of the periodic noise is then:(8)$$\begin{array}{cc} \omega & ∼\omega_{\mathrm{true}}+\sum_{m=0}^{N/2}A_{m}\cdot \sin\left(2\pi f_{m}t+\varphi_{m}\right)\\ & ∼\omega_{\mathrm{true}}+\sum_{m=0}^{N/2}A_{m}\cdot \sin\left(2\pi m\frac{F_{s}}{N}t+\varphi_{m}\right)\\ & ∼\omega_{\mathrm{true}}+\sum_{m=0}^{N/2}A_{m}\cdot \sin\left(2\pi \frac{m}{NT_{s}}t+\varphi_{m}\right), \end{array}$$
where *A* is the amplitude of the measured oscillations (not to be confused with the amplitude of the vibrations itself), *T*_s_ is the sampling period of the sensor, and *φ* is the phase of the measured oscillations. Altogether, the measured value at the output of the single-axis gyroscope is:(9)$$\omega_{\mathrm{raw}}=\Delta \omega \cdot \left\|\frac{\min\left(\max\left(G\omega_{\mathrm{true}}\cos\varepsilon +B+\nu +\sum_{m}A_{m}\cdot \sin\left(\frac{2\pi mt}{NT_{s}}+\varphi_{m}\right),\;-\omega_{\max}\right),\;\omega_{\max}\right)}{\Delta \omega }\right\|.$$

If we use the sensor with sufficient full-scale range $\pm \omega_{\max}$ and a very small quantization step Δω, the model of the sensor can be considered linear. The mean square error of the estimated angular velocity is:(10)$$\begin{array}{cc} \mathrm{MSE}(\omega ) & =\frac{\sum_{k=0}^{N-1}{\left(\omega_{\mathrm{raw}}[k]-\omega_{\mathrm{true}}[k]\right)}^{2}}{N}\\ & =\frac{1}{N}\sum_{k=0}^{N-1}{\left(G\omega_{\mathrm{true}}[k]\cos\varepsilon +B+\nu [k]-\omega_{\mathrm{true}}[k]+\sum_{m}A_{m}\cdot \sin\left(\frac{2\pi mt}{NT_{s}}+\varphi_{m}\right)\right)}^{2}\\ & \approx \frac{1}{2}\sum_{m}A_{m}{}^{2}+B^{2}+\frac{{\left(\gamma -1\right)}^{2}}{N}\sum_{k=0}^{N-1}\omega_{\mathrm{true}}{\left[k\right]}^{2}+\frac{1}{N}\sum_{k=0}^{N-1}\nu {\left[k\right]}^{2}, \end{array}$$
where γ=Gcosε is the directional gain of the sensor. Since we assume that the internal noise component ν[k] is independent from the true value $\omega_{\mathrm{true}}[k]$, the mean of their product is negligible. At zero rate (during static calibration, $\omega_{\mathrm{true}}[k]=0$), the output bias *B* can be observed at the output of the gyroscope as the mean value, and the MSE is:(11)$$\mathrm{MSE}(\omega_{0})=\frac{1}{2}\sum_{m}A_{m}{}^{2}+B^{2}+\frac{1}{N}\sum_{k=0}^{N-1}\nu {\left[k\right]}^{2}.$$

The same principles and formulas are valid for accelerometers or any sensors sensitive to the environmental noise.

### 2.3. Synchronization Errors

Within a multi-sensor environment, we need to consider the synchronization of individual sensors. Some intelligent sensors provide a trigger input, which allows the master controller (e.g., a microprocessor) to send a synchronization pulse, starting the measurement of all sensors at the same time. To quantify the impact of incorrect synchronization, we may assume two identical sensors with the same sampling period *T_s_* but with different sampling phases. Qualitatively, the effect of invalid synchronization increases when the measured variable changes rapidly. If we model the dynamic input signal (true measured angular velocity) as a sine wave with the frequency *f_sig_*, the readings of two sensors are:(12)$$\omega_{1}[k]=A\sin\left(2\pi f_{sig}t\right)+\nu =A\sin\left(2\pi \frac{f_{sig}}{F_{s}}k\right)+\nu ,$$
(13)$$\omega_{2}[k]=A\sin\left(2\pi f_{sig}\left(t-\tau \right)\right)+\nu =A\sin\left(2\pi \frac{f_{sig}}{F_{s}}k-2\pi f_{sig}\tau \right)+\nu ,$$
where *A* is the signal amplitude, ν is white Gaussian noise superposed independently to both sensors, and *τ* is the synchronization delay of the second sensor. Naïve sensor fusion is the average of the two sensors:(14)$$\omega [k]=0.5\omega_{1}[k]+0.5\omega_{2}[k].$$

Figure 2 shows the RMSE of the value estimated using equation (14 concerning the relative sampling frequency and synchronization delay. The RMSE of each sensor noise was set to 1% of the signal amplitude for the simulation. A significantly larger sampling frequency than signal frequency (20 times and more) suppresses the adverse effects of invalid sensor synchronization (note the logarithmic vertical scale).

## 3. Homogeneous Sensor Fusion

We may now analyze the situation when multiple almost-identical sensors sense the same variable. While the environmental noise (e.g., vibrations) is common for all sensors, the random internal sensor noise ν is purely random. The homogeneous sensor fusion can be expressed as a weighted average of the sensors’ readings:(15)$$\Omega [k]=\sum_{j}q_{j}\omega_{j(\mathrm{raw})}[k]\;\mathrm{and}\;\sum_{j}q_{j}=1,$$
where *q_j_* is the weight of the *j*-th sensor. If we apply Equations (9)–(15), neglecting the quantization and limits of the sensor, we obtain:(16)$$\begin{array}{l} \Omega [k]=\sum_{j}q_{j}\left(\gamma_{j}\omega_{\mathrm{true}}[k]+B_{j}+\sum_{m}A_{m}\cdot \sin\left(\pi m\frac{t}{T_{s}}+\varphi_{m}\right)+\nu_{j}[k]\right)\\ =\sum_{m}A_{m}\cdot \sin\left(\pi m\frac{t}{T_{s}}+\varphi_{m}\right)+\omega_{\mathrm{true}}[k]\cdot \sum_{j}q_{j}\gamma_{j}+\sum_{j}q_{j}B_{j}+\sum_{j}q_{j}\nu_{j}[k]. \end{array}$$

The goal of the sensor fusion is to minimize the MSE of the output (least-squares method). Theoretically, if the gain and bias errors are completely compensated by calibration, the sensor is perfectly aligned, and there are no vibrations present, the only source of the error is the internal noise of the sensor, and the result of the sensor fusion is:(17)$$\Omega_{\mathrm{ideal}}[k]=\omega_{\mathrm{true}}[k]+\sum_{j}q_{j}\nu_{j}[k],$$

The MSE of the sensor fusion is then:(18)$$\mathrm{MSE}(\Omega_{\mathrm{ideal}})=\sum_{j}q_{j}{}^{2}\mathrm{MSE}(\omega_{j}).$$

When each ideal sensor has a weight coefficient proportional to $q_{j}∼1/\mathrm{MSE}(\omega_{j})$, the MSE of the ideal sensor fusion result MSE(Ω_ideal_) will be minimal (see, e.g., [28]).
(19)$$\mathrm{MSE}{\left(\Omega_{\mathrm{ideal}}\right)}_{\min}=\frac{\sum_{j}\frac{1}{\mathrm{MSE}(\omega_{j})}}{{\left(\sum_{j}\frac{1}{\mathrm{MSE}(\omega_{j})}\right)}^{2}}=\frac{1}{\sum_{j}\frac{1}{\mathrm{MSE}(\omega_{j})}}.$$

The precision of the real fusion result depends on the precise estimation of the MSE. If we use the zero-rate estimate of the MSE (Equation (11)), which is an overestimate, the MSE of the sensor fusion result will be worse. We do not know the harmonic components of the measured vibrations (parameters *A_m_*, *φ_m_*), nor the values of the calibration parameters after a longer run. The values of the calibration parameters for a group of sensors are considered randomly distributed around their ideal values. When the homogeneous sensor fusion (weighted average) provides a reasonable estimate of the true value, it is possible to estimate the values of the calibration parameters from the last *N* samples. The simple calibration model is:(20)$$\omega_{j(\mathrm{cal})}[k]=\left(1+c_{1j}\right)\cdot \omega_{j(\mathrm{raw})}[k]+c_{2j},$$
where $\omega_{j(\mathrm{raw})}[k]$ is the *k*-th raw digital sample of the *j*-th sensor converted to SI units, $\omega_{j(\mathrm{cal})}[k]$ is the corresponding calibrated value, and $c_{1j},c_{2j}$ are the calibration parameters. The Equation (20) represents the first-order (linear) calibration model. The sensor gain should ideally be equal to one; the *c*_1_ parameter represents gain deviation and is usually significantly lower than one. The form of the calibration model was chosen to improve numerical precision when floating point numbers are used. The calibration parameters *c*_1_ and *c*_2_ may be considered quasi-constant since they may slightly vary with the elapsed time and temperature. In the ideal case, both calibration parameters are zero. If we invert (20) and compare it with (7), the gain and bias of *j*-th sensor *G_j_*, *B_j_* from (7) can be rewritten in terms of *c*_1*j*_, *c*_2*j*_:(21)$$G_{j}=\frac{1}{1+c_{1j}},$$
(22)$$B_{j}=\frac{-c_{2j}}{1+c_{1j}},$$

If we would know the true value and the noise is Gaussian, the calibration parameters can be estimated using the least-squares method:(23)$$\underset{y_{j}}{\underset{︸}{\left(\begin{array}{c} \omega_{\mathrm{true}}[1]-\omega_{j\left(\mathrm{raw}\right)}[1]\\ \omega_{\mathrm{true}}[2]-\omega_{j\left(\mathrm{raw}\right)}[2]\\ \ldots \\ \omega_{\mathrm{true}}[K]-\omega_{j\left(\mathrm{raw}\right)}[K] \end{array}\right)}}≃\underset{W_{j}}{\underset{︸}{\left(\begin{array}{cc} \omega_{j\left(\mathrm{raw}\right)}[1] & 1\\ \omega_{j\left(\mathrm{raw}\right)}[2] & 1\\ \ldots & 1\\ \omega_{j\left(\mathrm{raw}\right)}[K] & 1 \end{array}\right)}}\cdot \left(\begin{array}{c} c_{1j}\\ c_{2j} \end{array}\right),$$
(24)$$\left(\begin{array}{c} c_{1j}\\ c_{2j} \end{array}\right)={\left(W_{j}{}^{T}\cdot W_{j}\right)}^{-1}\cdot W_{j}{}^{T}\cdot y_{j},$$

In a real application, the true value is unknown. In the case of multiple sensors, the apriori estimate of the true value is the mean of all calibrated sensors’ measurements at the given time (all *q_j_* are the same):(25)$$\Omega_{est}[k]=\frac{1}{M}\sum_{j}\omega_{j\left(\mathrm{cal}\right)}[k],$$
where *M* is the count of sensors. Initially, the bias and gain deviation may be set to zero; hence, $\omega_{j\left(\mathrm{cal}\right)}[0]=\frac{1}{M}\sum_{j}\omega_{j\left(\mathrm{raw}\right)}[0]$.

Algorithm 1 explains the homogeneous sensor fusion with the calibration estimation:
**Algorithm 1** Homogeneous sensor fusion with calibrationset: c1j←0, c2j←0, qj←1M**for** *j* = 1 **to** *M*  get ωj(raw)**end for**initialize the estimate of the true value: Ωest←1M∑jωj(raw)compute sensor deviation yj←Ωest−ωj(raw)compute the calibration parameters *c*_1_, *c*_2_ using (24)**for** *j* = 1 **to** *M*  compute calibrated values ωj(cal)←(1+c1j)⋅ωj(raw)+c2j**end for****for** r = 1 to iterations  re-compute the estimate of the true value: Ωest←∑jqjωj(cal)  **for** *j* = 1 **to** *M*    compute the MSE of each sensor MSE(ωj)←Var(Ωest−ωj(cal))    compute the weight of the sensor qj←1/MSE(ωj)  **end for**  compute maximal weight qmax=μM⋅∑jqj, truncation factor *µ* = 3 (empirical)  **for** *j* = 1 **to** *M*    truncate qj←min(qj,qmax)    normalize: qj←qj/∑kqk  **end for****end for**

In the real-time version of the above algorithm, we process a fixed-size window of historical raw readings. 

### Truncation Factor

The proposed algorithm has intrinsic instability. When the MSE of one sensor is significantly underestimated compared to the other sensors, the estimated MSE of that sensor is low and its weight *q_j_* is significantly higher than the weights of the other sensors. Such a sensor becomes a “dictator”—in the next iteration, the estimated weighted average $\Omega_{\mathrm{est}}$ is pulled towards the readings of the dictator sensor. Therefore, it is pulled away from the readings of the other sensors. Then, the algorithm overestimates the MSE of the other sensors, resulting in a further decrease in their weight. After multiple iterations, the normalized weight of the dictator sensor converges to one, and all other sensors are suppressed. To avoid such behavior, truncation factor *µ* was introduced. It ensures that the weight of the best sensor never exceeds *µ*-times the average weight of all sensors. Suppose we assume that the RMS of a single sensor is a gamma-distributed random variable. In that case, the probability of false truncation (underestimation of the best sensor) is shown in Figure 3. The parameter shape *θ* = 23 ± 8 (std. deviation) and scale *β* = 0.03 ± 0.01 of the gamma distribution were roughly estimated by fitting the histogram of the RMS values obtained from a sample of 16 sensors. Figure 4 shows the RMSE of the estimated angular velocity concerning the truncation factor (see the next section for further details about the simulation parameters). According to the simulation results, the optimal truncation factor is approximately 3, which is the value to be used for further experiments. The false sensor weight truncation probability for *μ* = 3 is around 1%.

## 4. Simulation

We have used both simulation and real experiments to test the proposed method’s performance. The simulation raw data were computed using MATLAB R2021a environment with the following parameters:count of the samples *N* = 10,000sampling frequency *F*_S_ = 100 Hzamplitude of the random signal *A* = 200 deg·s^−1^frequency vector of the random signal *f*[*k*] = 1.0 Hz + *ξ*, where *ξ* is normally distributed random number (implemented by MATLAB function randn)phase of the random signal $\varphi [k]=\frac{2\pi }{F_{S}}\sum_{n=1}^{k}f[n]$ (implemented as MATLAB function cumsum)simulated (true) angular velocity: $\omega_{\mathrm{true}}[k]=A\cdot \sin\left(\varphi [k]\right)$RMS of all 16 sensors is from a gamma distribution with shape α = 5 and scale β = 0.02bias of all 16 sensors is from a zero-centered Gaussian distribution with standard deviation *σ* = 30 deg·s^−1^gain of all 16 sensors is one-centered with standard deviation *σ* = 0.04

The above parameters roughly correspond to the noise parameters of real low-cost MEMS gyroscopes available commercially. Using a pseudo-sinusoidal signal with a fluctuating frequency simulates the continuous angular velocity of a physical object with a non-zero moment of inertia. An example of the simulated signal is in Figure 5. The average gain across all sensors equals one and the average bias is zero. Such normalization is necessary because the algorithm cannot intuitively determine the common-cause bias of all sensors without any apriori information (e.g., from different types of sensors). For simplicity, when all sensors deviate in the same direction, the result will also deviate in that direction.

## 5. Simulation Results

In the simulation mode, each sensor’s gain, bias, and MSE are known. Therefore, we may compute the optimal weights and compare the estimated parameters with the known values (see Table 1).

The algorithm converges under the abovementioned ideal conditions (zero-centered biases, one-centered gains). The shown values are obtained by three iterations of the RMS estimation. Please note that the algorithm, as described, computes MSE (mean square error) instead of RMS to avoid unnecessary computation of the square root.

## 6. Experimental Setup

Real-world experiments were conducted using a matrix of 16 MPU9250 MEMS sensors from TDK InvenSense (Shenzhen, China) (see Figure 6). All sensors communicate with the microcontroller STM32F446 from STMicroelectronics (Geneva, Switzerland) via two independent SPI channels. The microcontroller then sends all sensor readings to the computer via serial connection. To synchronize the sensor readings, all sensors contain synchronization trigger input. The synchronized readings are stored within internal FIFO buffers within each sensor and later retrieved via SPI. The typical noise characteristics of the used gyroscope sensor obtained from the datasheet are [29]:gain tolerance ±3% (at 25 °C), ±4% (whole temperature range −40 °C to +85 °C)nonlinearity ±0.1%bias tolerance ±5 deg·s^−1^ (at 25 °C), ±30 deg·s^−1^ (whole temperature range from −40 °C to +85 °C)RMS noise 0.1 deg·s^−1^

To evaluate the proposed method, it is necessary to measure the ground-truth value of the angular velocity. Our first attempts utilized a stepper motor driving a rotational platform. Still, the MEMS gyroscopes mentioned above captured the high-frequency changes in the motor’s rotation speed caused by the steps. The frequency of those steps is higher than the Nyquist frequency of the sensors, thus causing aliasing. To circumvent that, we attached the sensor matrix to a commercially available SPAN-CPT inertial navigation unit from NovTel Inc. (Calgary, AB, Canada), which contains three DSP-3000 single axial fiber optical gyroscopes (FOGs) manufactured by KVH Industries (Middletown, CT, USA). The characteristics of FOG sensors are [30]:sampling frequency: 1000 Hz (downsampled by SPAN-CPT to 100 Hz)initial bias: ±20°/hnonlinearity: 500 ppm at ±150°/hbias stability: 1°/hangle random walk: 0.067°/√hRMS (measured): 0.005°/s

It is clear that FOG has 20 times lower RMS, so it thus may be used as a source of ground-truth value. Initial calibration allows us to compensate for the bias of the FOG sensor. The matrix of MEMS sensors attached to the SPAN-CPT unit is shown in Figure 7. Both sensors were rotated randomly by hand. The time series of the measured angular velocity from the MEMS array and FOG was roughly synchronized using a timestamp and fine-aligned by hand to suppress delays caused by the communication interfaces in the PC. The manual alignment of MEMS and FOG signals is needed only for validation purposes; in real applications, the FOG is not present. The measured angular velocity is in Figure 8.

The estimated gain, bias, and weight of all sensors are shown in Figure 9, which presents a comparison of the sensor calibration parameters. Blue circles are “true” values considering the FOG data. The results are obtained using a moving window with the size of 1000 samples (processing the last 10 s). Shown outliers represent short-term calibration changes.

As seen in Table 2 and Figure 9, the bias and gain of sensors were estimated correctly. The true gain and bias were obtained by comparing the results of the MEMS sensor with the readings from FOG. Those reference values lay within bounding boxes of estimated gain and bias. The mean absolute error of the gain estimation was 0.46%, and the mean absolute error of the bias estimation was 0.04 deg·s^−1^ (0.016% of the gyroscope full scale). The weights of the sensors are estimated roughly, with a mean absolute error of 41%, but it reflects the qualitative difference between the sensors (e.g., sensors *j* = 3, 4, 14, 15 have low weights). Table 3 compares two scenarios: the first with the real measured data and the second with an artificial white Gaussian noise added to one of the sensors, which emulates the degradation.

## 7. Conclusions

This article deals with the real-time calibration and sensor fusion in the homogeneous sensor array of the MEMS gyroscopes, measuring the angular velocity. The proposed method has been validated by both simulation and real-world experiments. The results of the experiments show that the algorithm improves the precision of the estimated angular velocity by approx. 5% compared to a naïve average when all sensors are working. The main advantage of the algorithm is the intrinsic ability to compensate for the degradation of some sensors using an adaptive weighted average, thus improving the reliability of the sensor array. The homogeneous sensor fusion can estimate calibration parameters (gain, bias) of individual sensors within the sensor array without the need for precise offline calibration equipment. Compared to the LVM (linear minimum variance) methods, our method does not require apriori information about error covariance matrices for individual sensors and can capture the changes in the error characteristics in real time. The proposed method applies to all types of sensors (not exclusively MEMS gyroscopes) because it does not require additional information. One of the critical challenges in such a configuration is to keep sensors in sync, which can be achieved by a global synchronization signal routed to all sensors. This drawback can be neglected when the sampling period is significantly smaller than the time constant of the system dynamics.

## Figures and Tables

**Figure 1 sensors-23-06431-f001:**
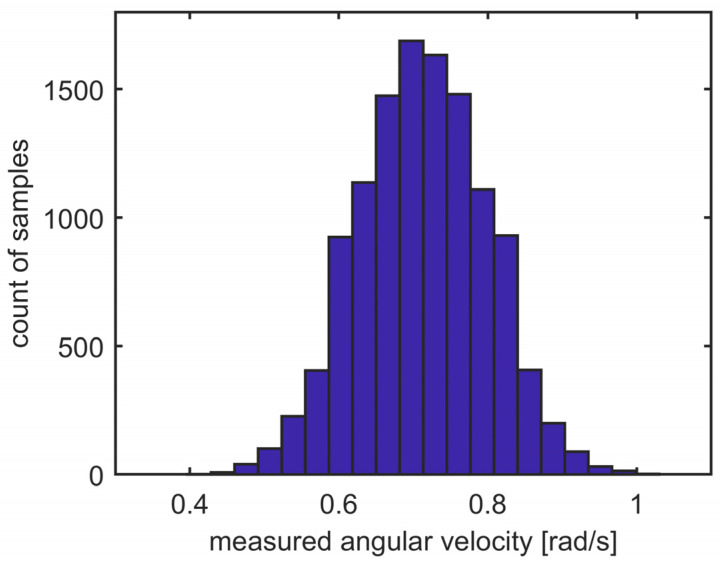
Histogram of the sensor noise.

**Figure 2 sensors-23-06431-f002:**
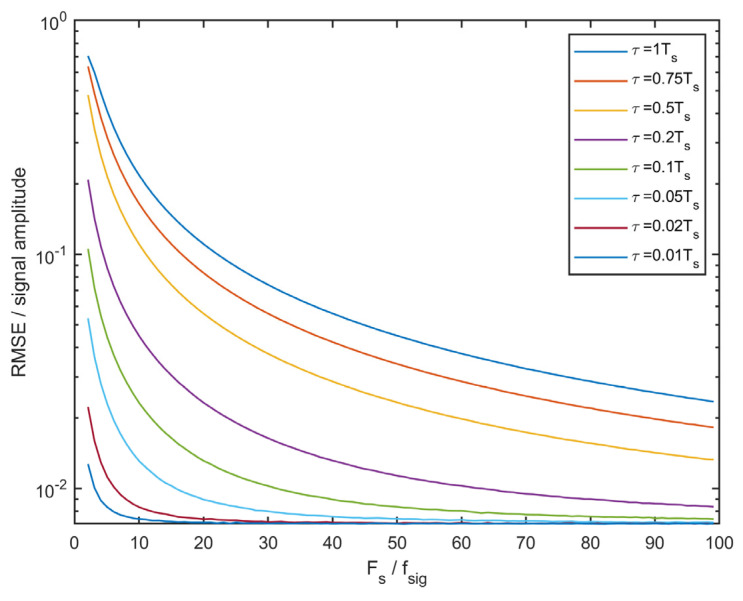
RMSE of the sensor fusion concerning the sampling frequency and synchronization delay between two identical sensors.

**Figure 3 sensors-23-06431-f003:**
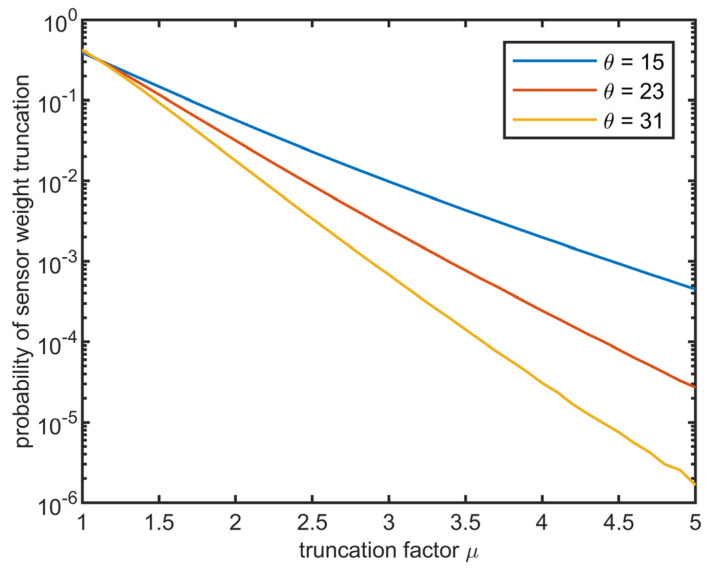
Simulated probability of the false sensor weight truncation using the scale factor of the gamma distribution with the shape *θ* = 23 ± 8 and scale *β* = 0.82/*θ*.

**Figure 4 sensors-23-06431-f004:**
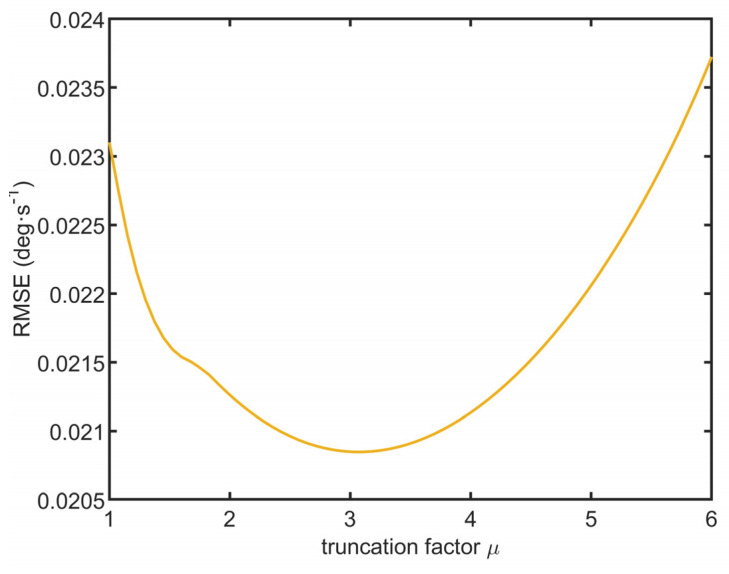
RMS error of the estimated angular velocity vs. truncation factor.

**Figure 5 sensors-23-06431-f005:**
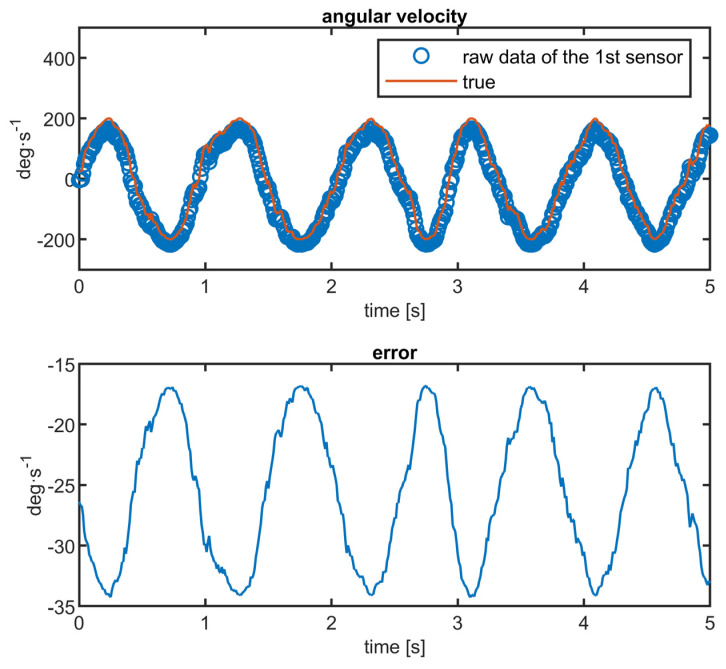
Simulated angular velocity.

**Figure 6 sensors-23-06431-f006:**
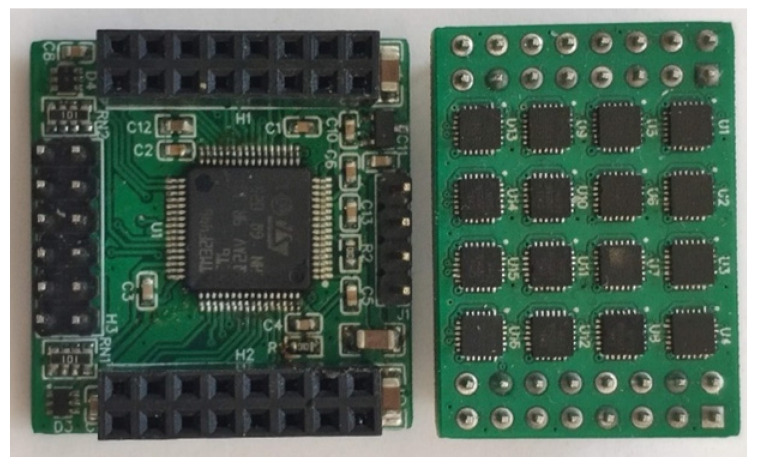
Sensor matrix (**right**) and microcontroller module (**left**).

**Figure 7 sensors-23-06431-f007:**
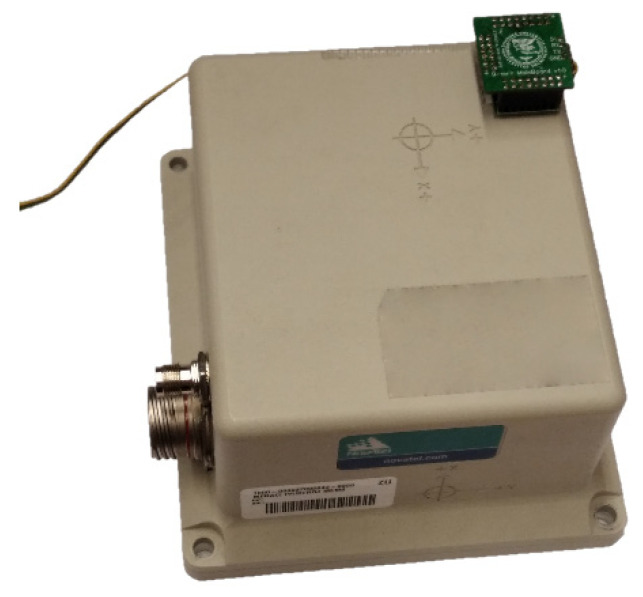
MEMS sensor array mounted at the top of the FOG unit.

**Figure 8 sensors-23-06431-f008:**
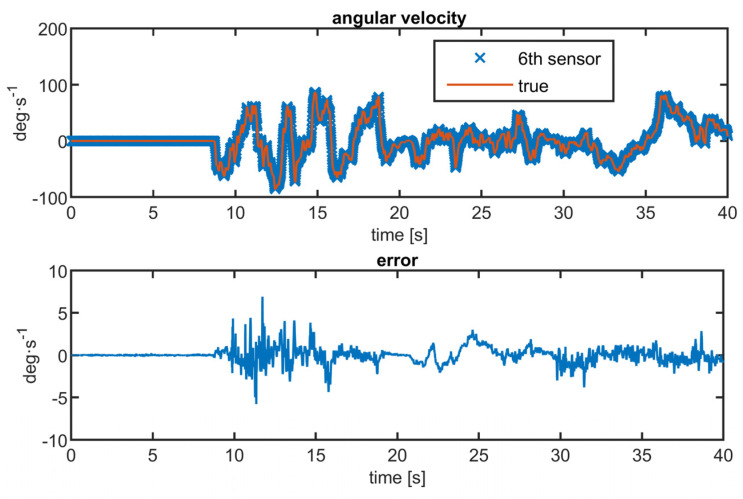
Comparison between 6th MEMS gyroscope and true value from FOG.

**Figure 9 sensors-23-06431-f009:**
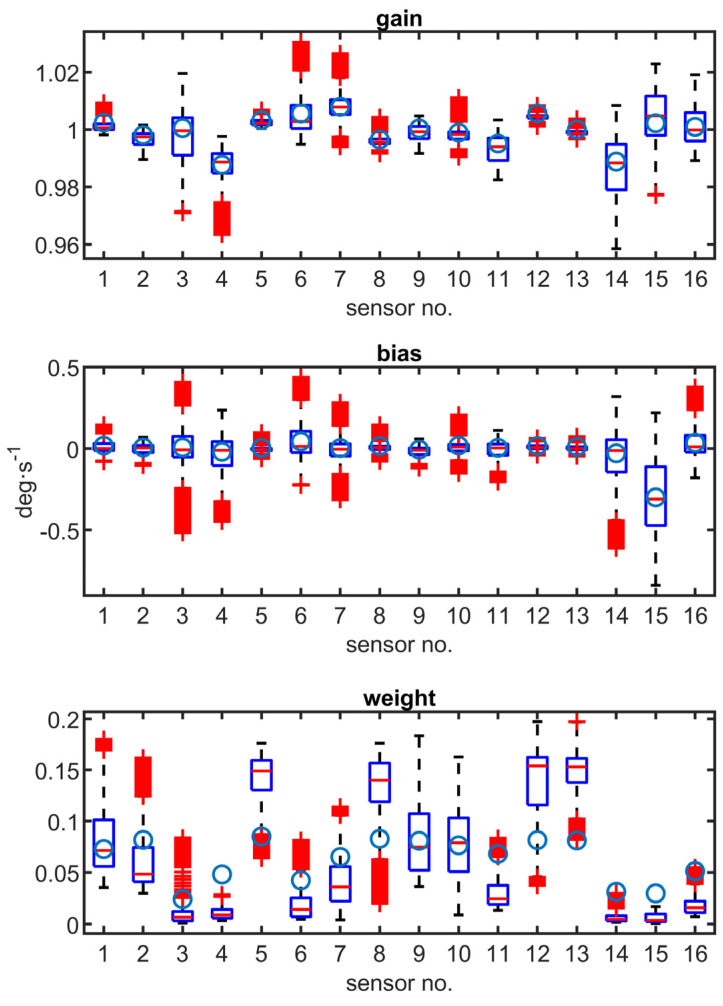
Comparison of sensors’ calibration parameters. Blue circles are “true” values considering the FOG data.

**Table 1 sensors-23-06431-t001:** Convergence of simulation.

Sensor No. *j*	Gain *G_j_* (-)		Bias *B_j_* (deg·s^−1^)		RMS*_j_* (deg·s^−1^)	
	Estimated	True	Estimated	True	Estimated	True
1	0.956946	0.956949	−25.5111	−25.5107	0.103	0.101
2	0.998366	0.998368	−8.0970	−8.0975	0.097	0.099
3	0.963520	0.963516	−13.9765	−13.9773	0.077	0.077
4	1.016634	1.016636	6.7336	6.7339	0.085	0.088
5	0.990664	0.990654	34.6136	34.6141	0.163	0.162
6	0.963372	0.963376	−6.1223	−6.1218	0.049	0.053
7	0.990176	0.990172	46.9899	46.9911	0.147	0.147
8	0.987289	0.987301	15.0531	15.0539	0.101	0.102
9	1.008443	1.008421	−81.3185	−81.3222	0.198	0.204
10	1.053295	1.053302	−23.8845	−23.8849	0.094	0.101
11	1.003599	1.003604	45.7136	45.7147	0.076	0.079
12	0.996136	0.996143	−33.4840	−33.4843	0.055	0.060
13	1.056180	1.056180	11.0308	11.0311	0.064	0.070
14	0.988030	0.988037	−0.2502	−0.2506	0.123	0.122
15	0.972770	0.972752	2.3391	2.3411	0.177	0.173
16	1.054589	1.054589	30.1703	30.1694	0.072	0.078

**Table 2 sensors-23-06431-t002:** Estimated parameters of individual sensors.

Sensor No. *j*	Gain *G_j_* (-)		Bias *B_j_* (deg·s^−1^)		Weight *q_j_* (-)	
	Estimated	True	Estimated	True	Estimated	Optimal
1	0.9986	1.0024	−0.0026	0.0176	0.0580	0.0726
2	1.0007	0.9981	0.0044	0.0064	0.0407	0.0816
3	1.0019	1.0005	−0.0092	0.0065	0.0836	0.0241
4	0.9957	0.9878	−0.0110	−0.0195	0.0149	0.0480
5	1.0018	1.0033	−0.0068	0.0038	0.0912	0.0849
6	0.9957	1.0056	0.0132	0.0437	0.0808	0.0422
7	1.0106	1.0079	−0.0042	0.0028	0.1075	0.0649
8	0.9956	0.9965	0.0134	0.0160	0.1190	0.0827
9	1.0022	1.0004	−0.0096	−0.0064	0.0850	0.0807
10	0.9991	0.9989	0.0115	0.0183	0.0561	0.0762
11	0.9975	0.9951	0.0120	0.0044	0.0828	0.0679
12	1.0041	1.0056	−0.0011	0.0161	0.0377	0.0813
13	0.9981	1.0000	−0.0001	0.0102	0.0820	0.0810
14	0.9999	0.9888	−0.0125	−0.0289	0.0291	0.0311
15	1.0172	1.0022	−0.7651	−0.2994	0.0021	0.0297
16	0.9927	1.0009	0.0098	0.0349	0.0297	0.0511

**Table 3 sensors-23-06431-t003:** Comparison of results.

Constellation	RMSE (deg·s^−1^)	
	Measured Data	Single Sensor Additive Noise
Single sensor	0.945	10.000
The mean of 16 sensors, no calibration	0.720	0.936
The mean of 16 sensors, with calibration	0.701	0.924
The weighted sum of 16 sensors, with calibration	0.685	0.688

## Data Availability

All data associated with the article are available online at https://github.com/dusan-nemec/mems-calib (accessed on 10 May 2023).

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
