# Peer review of "Homogeneous Sensor Fusion Optimization for Low-Cost Inertial Sensors"

_sensors, 2023, doi:10.3390/s23146431_

Round 1
Reviewer 1 Report
please see the attached file

Can be improved.
Author Response
Q1: Specific explanation of equation (3) should be shown.
A1: The explanation of symbols for the eq. (3) has been added.
--------------------------------
Q2: What is the theoretical basis for the modeling of equation (8)? Why amplitude,
frequency and phase of vibration can be equivalent to amplitude, frequency and
phase of gyro output signal component.
A2: The equation (8) does not contain the amplitude, frequency and phase of the vibrations itself, but the properties measured by the sensor. The sensitivity of the gyroscope to the vibrations varies with frequency and direction of the vibrations. Equation (8) and related text has been rewritten to avoid confusion.
-------------------------------
Q3: .. So what is the relationship between systematic errors and the fusion?
A3: Values of calibration parameters for a group of sensors are considered randomly distributed around their ideal values. When the homogeneous sensor fusion (weighted average) provides good estimate of the true value, it is possible to estimate the values of the calibration parameters from last N samples. Because the weighted average is computed from the calibrated values, the important contribution of the article is related to the stability of the algorithm. Explanation has been added to the paper.
--------------------------------
Q4: What is the theoretical basis for the modeling of equation (17)? Specific
explanation should be given.
A4: The equation represents standard linear calibration model, but is rewritten into form which is more numerically stable when implemented using floating-point numbers.
Explanation for the equation in the paper has been extended to avoid confusion.
--------------------------------
Q5: SPIs collecting 16 sensors will cause sampling delay, how do you consider the
error caused by this non-synchronous sampling? And how do FOG and sensor
array output synchronize completely in comparison.
A5: In order to synchronize the sensor readings, all sensors contain synchronization trigger input. It allows to capture synchronized readings, which are then stored within internal FIFO buffers within each sensor and later retrieved via SPI.
The time series of the measured angular velocity from MEMS array and from FOG was roughly synchronized using timestamp and fine aligned by hand to suppress delays caused by communication interfaces in PC.
----------------------------------
Q6: The article mentions “Obviously, we do not know the harmonic components of
the vibrations (parameters Am, φm), nor the calibration parameters.”. However,
in equations (17)-(19), the relationship between calibration parameters c1 and c2
with gain Gj and bias Bj appears. Please explain the basis for the relationship
between c1, c2 and Gj, Bj in equations (18), (19). And please explain the basis
for the determination of normalize gain and normalize bias in Algorithm 1.
A6: We know only the initial values of the calibration parameters, but they change over time. Clarification has been introduced into paper. The relation between c1j, c2j and Gj, Bj can be obtained when inverting eq. (17) and comparing with (7), which is a trivial operation. The subscript {j} was added to keep consistend notation. The gain and bias normalization was used only in simulation, and was removed from the final version of the algorithm. The text of the article has been corrected accordingly.
-------------------
Reviewer 2 Report
Comments are divided into two categories: minor and major. Minor edits are highlighted in the attached file for authors' convenience/reference.
Minor edits:
page 1, ln 32: "changes" should be "change" for subject-verb agreement
page 1, ln 38: "calibration which" should be "calibration, which"
page 2, ln 47: "RANSAC" not defined
page 2. ln 54: "to compensate" would read better as "compensation"
page 2, ln 55: "FIR" not defined
page 2, ln 68: "allowing to claibrate the sensors the mutual sensor" does not make sense. Please reword for clarity
page 4, ln 132: "varies" should be "vary" for subject-verb agreement
page 6, ln 173: qj needs the j as subscript
Overall, the paper needs substantial English work for readability, specifically definite/indefinite article usage.
Major edits:
1) The English language was rough. There was a significant lack of definite and indefinite articles ("the" and "a/an") that made the text awkward and confusing.
2) Figure 2 could benefit from a range of gamma shapes (i.e., a range of theta). The authors do not indicate why 10 was chosen, nor do they indicate whether or not 10 is significant in any way. If it's an arbitrary choice for example, then a range of example values would be better.
3) The authors chose a truncation value of mu=3. Why? What would be the implications of other values? When would they be appropriate, if ever?
3) Figure 7 would benefit from expanded explanation of what the reader is seeing and how it should be interpreted. Page 12, ln 278-279 says "the bias and gain of sensors were estimated correctly. The weight of the sensors are estimated roughly, but it reflects the qualitative difference between the sensors." How does the reader know this is true from the results presented?
4) Table 2: Quantitatively, the proposed method produces better results. What is the practical advantage in the improvement that is shown?
5) There were a lot of assumptions presented to produce the equations used in the presented algorithm. Did the authors do any analysis of the experimental data to show the assumptions were valid? Some explanation of this aspect would be appropriate.

See comments in "suggestions for authors"
Author Response
Minor edits were implemented as suggested in the attached document. We thank the reviewer for his/her precise and constructive comments.
-----------------------------------------
Q1: The English language was rough. There was a significant lack of definite and indefinite articles ("the" and "a/an") that made the text awkward and confusing.
A1: Missing articles have been corrected.
---------------------------------------
Q2: Figure 2 could benefit from a range of gamma shapes (i.e., a range of theta). The authors do not indicate why 10 was chosen, nor do they indicate whether or not 10 is significant in any way. If it's an arbitrary choice for example, then a range of example values would be better.
A2: We have roughly estimated the parameters of gamma distribution from the measured RMS values of the sensors within sensor array. The figure 2 has been updated accordingly to reflect the estimated shape parameter.
---------------------------------------
Q3: The authors chose a truncation value of mu=3. Why? What would be the implications of other values? When would they be appropriate, if ever?
A3: The value of the truncation factor was set empirically, based on simulation. Figure 3 showing the dependency of the estimation error on the truncation factor, has been added to the paper to clarify the choice for mu value.
--------------------------------------------
Q4: Figure 7 would benefit from expanded explanation of what the reader is seeing and how it should be interpreted. Page 12, ln 278-279 says "the bias and gain of sensors were estimated correctly. The weight of the sensors are estimated roughly, but it reflects the qualitative difference between the sensors." How does the reader know this is true from the results presented?
A4: The evaluation using mean absolute error of the estimated parameters (gain, bias and weight) including more explanation was added to the paper.
-----------------------------------------------
Q5 Table 2: Quantitatively, the proposed method produces better results. What is the practical advantage in the improvement that is shown?
A5: Main advantage of the proposed method is the intrinsic ability to compensate degradation of some sensors. It also allows to estimate the calibration parameters (gain, bias) without need for precise calibration equipment and also to capture the changes of those parameters.
----------------------------------
Q6: There were a lot of assumptions presented to produce the equations used in the presented algorithm. Did the authors do any analysis of the experimental data to show the assumptions were valid? Some explanation of this aspect would be appropriate.
A6: We assume that the confusion has been caused mainly by the paragraph about vibrations and their impact on sensor readings. The paragraph has been rewritten to clarify that we do not assume anything about the vibrations itself, only that it causes periodic noise, which can always be represented as a sum of harmonic components in discrete time. We also assume that the random sensor noise is Gaussian, which is clarly demonstrated by Figure 1.
Round 2
Reviewer 1 Report
The previous comments are responded in this revision. Here, some additional questions still need to be replied. 1). Please explain Eq. (3), (8) and (17). 2) synchronization trigger input used in this paper is a useful method to achieve multi-sensor synchronous sampling. “fine aligned by hand to suppress delays” used in this work maybe useful in low dynamic scenarios, How to avoid non-synchronous sampling error to achieve the good effect in high dynamic application. 3) Please state the difference between the fusion method mentioned in this work and traditional fusion methods such as minimum variance fusion.. 4) It is mentioned in the conclusion that the sensor synchronization error can be ignored when the sampling period is significantly smaller than the time constant of the system dynamics. Is there any theoretical support for this conclusion? If so, please add relevant references or theoretical derivation.
Moderate editing of English language required
Author Response
Thanks again for valuable comments and suggestions.
------------------------------------------
Q1) Please explain Eq. (3), (8) and (17).
A1) Explanation for the mentioned equations has been added to the paper.
-----------------------------------------
Q2) synchronization trigger input used in this paper is a useful method to achieve multi-sensor synchronous sampling. “fine aligned by hand to suppress delays” used in this work maybe useful in low dynamic scenarios, How to avoid non-synchronous sampling error to achieve the good effect in high dynamic application.
A2) The manual alignment of MEMS and FOG signals is needed only to validate the MEMS array, in real application the FOG is not present. During experiments we may control the shape of the signal, with multiple zero-rate periods.
-----------------------------------------
Q3) Please state the difference between the fusion method mentioned in this work and traditional fusion methods such as minimum variance fusion..
A3) Compared to the LVM (linear minimum variance) methods our method does not require apriori information about error covariance matrices for individual sensors. The explanation has been added to the paper.
-----------------------------------------
Q4) It is mentioned in the conclusion that the sensor synchronization error can be ignored when the sampling period is significantly smaller than the time constant of the system dynamics. Is there any theoretical support for this conclusion? If so, please add relevant references or theoretical derivation.
A4) Section 2.3 has been added to the article to better explain the effects of sychnronization error with respect to the sampling frequency.
Reviewer 2 Report
Thank you for addressing my concerns.
Author Response
Thanks again for your previous valuable comments and suggestions.